# Prognostic Factors Associated with Tumor Recurrence and Overall Survival in Soft Tissue Sarcomas of the Extremities in a Colombian Reference Cancer Center

Sandra E. Díaz Casas [1,*], Juanita Martínez Villacrés [2], Carlos Lehmann Mosquera [1], Mauricio García Mora [1], Iván Mariño Lozano [1], Javier Ángel Aristizábal [1], Raúl Suarez Rodríguez [1], Carlos Alfonso Duarte Torres [1] and Ricardo Sánchez Pedraza [3]

[1] Functional Unit for Breast and Soft Tissue Tumors, Instituto Nacional de Cancerología, Bogotá 111511, Colombia; jangel@cancer.gov.co (J.Á.A.); rasuarez@cancer.gov.co (R.S.R.); cduarte@cancer.gov.co (C.A.D.T.)
[2] Fundación Universitaria de Ciencias de la Salud, Bogotá 111411, Colombia
[3] Instituto Nacional de Cancerología (INC)—Empresa Social del Estado, Universidad Nacional de Colombia, Bogotá 111511, Colombia; rsanchezpe@unal.edu.co
* Correspondence: sdiaz@cancer.gov.co

**Abstract: Introduction:** Soft tissue sarcomas (STS) are low-incidence tumors whose clinical and histopathological factors are associated with adverse oncological outcomes. This study evaluated prognostic factors (PF) associated with tumor recurrence and overall survival (OS) in patients diagnosed with STS of the extremities, treated at the Instituto Nacional de Cancerología (INC), Bogotá, Colombia. **Materials and Methods:** An analytical observational study of a historical cohort was carried out, including patients diagnosed with STS and managed surgically in the Functional Unit for Breast and Soft Tissue Tumors of the INC from January 2008 to December 2018. **Results:** A total of 227 patients were included; 74.5% had tumors greater than 5 cm. Most patients (29.1%) were in stage IIIB at diagnosis. Age was associated with higher mortality (HR = 1.01; CI95%: 1–1.02; $p = 0.048$). Tumor persistence at admission to the INC (HR = 2.34; CI95%: 1.25–4.35; $p = 0.007$) and histologic grade III (HR = 5.36; CI95%: 2.29–12.56; $p = <0.001$) showed statistical significance in the multivariate analysis for recurrence of any type, as did the PFs associated with a higher risk of local recurrence (HR = 2.85; CI95%: 1.23–6.57; $p = 0.014$ and HR = 6.09; CI95%: 2.03–18.2; $p = 0.001$), respectively. Tumor size (HR = 1.03; CI95%: 1–1.06; $p = 0.015$) and histologic grade III (HR = 4.53; CI95%: 1.42–14.49; $p = 0.011$) were associated with a higher risk of distant recurrence. **Conclusions:** This cohort showed that in addition to histologic grade and tumor size, tumor persistence at the time of admission has an impact on disease recurrence, so STS should be managed by a multidisciplinary team with experience in this pathology in high-volume reference centers.

**Keywords:** sarcoma; soft tissues; extremities; treatment

## 1. Introduction

Soft tissue sarcomas (STS) are a rare type of cancer, accounting for 1% of malignant tumors in adults [1]. According to data from the American Cancer Society, by 2022, 13,190 new cases were expected in the United States, with an estimated mortality of 5130 cases from this cause [2]. In Colombia, according to data from the statistical yearbook of the Instituto Nacional de Cancerología (INC), in 2020, there were 72 new cases found in the extremities and 40 in the retroperitoneum [3]. According to records of the database of the Functional Unit for Breast and Soft Tissue Tumors at the INC from July 2020 to July 2022, 125 new cases of STS were reported, 62 of them located in the extremities.

Although most sarcomas arise de novo, some risk factors have been identified for their appearance, such as exposure to radiation; environmental exposure to hydrochlorides

and herbicides; consumption of immunosuppressive and antineoplastic drugs; chronic lymphedema; infection by Herpes virus type 8 and Epstein–Barr virus; and hereditary syndromes, such as Gardner syndrome, Li–Fraumeni syndrome, neurofibromatosis type 1, Bloom syndrome, Werner syndrome, Rothmund–Thomson syndrome, familial adenomatous polyposis, among others [4,5].

Location is a factor influencing cancer treatment and oncological outcomes. Most STS are in the extremities (43%), trunk (10%), intra-abdominal area (19%), and retroperitoneum (15%) [4].

At present, there are more than 100 histologic subtypes [5], each of them with variable clinical behavior and presentation according to age. In children, the most common is rhabdomyosarcoma; in young adults, synovial sarcoma; and in the elderly, undifferentiated pleomorphic sarcoma (formerly malignant fibrous histiocytoma). Lymph node metastases are rare; however, they occur more frequently in epithelioid sarcoma, rhabdomyosarcoma, clear cell sarcoma, synovial sarcoma, and angiosarcoma [6]. The most common histologic types in the extremities are undifferentiated pleomorphic sarcoma, liposarcoma, leiomyosarcoma, synovial sarcoma, and malignant peripheral nerve sheath tumors [7].

Staging is performed using the AJCC eighth edition system, which classifies STS according to tumor size (T), lymph node involvement (N), presence of metastasis (M), and histologic grade (G). It is most often estimated according to the French system (Fédération Nationale des Centres de Lutte Contre le Cancer, FNCLCC), which has demonstrated a greater ability to predict oncological outcomes. This system includes tumor differentiation, mitotic count, and necrosis [8,9].

The prognosis of these tumor types is variable, and it depends on clinicopathological factors, the main one being histologic grade, followed by tumor size, depth of the lesion, histologic type, proximal location, state of the resection margins, and the patient's age, among others [5].

STS have high local recurrence rates that can reach up to 50% at 5 years and a 5-year overall survival ranging from 12% to 70%, depending on location and histologic type [10].

Treatment should be multidisciplinary in high-volume centers, which has shown a significant impact on the prognosis and survival of these patients [11]. Surgery is the mainstay of treatment, and its main objective is to achieve negative oncological margins to reduce the risk of local recurrence and, therefore, positively impact overall survival [5]. For decades, amputation was the most accepted surgical intervention in managing sarcomas of the extremities. Rosenberg et al. [12] demonstrated how limb-sparing surgery followed by radiotherapy was equivalent to radical surgery in terms of overall survival, with adequate local control. For this reason, limb-sparing procedures are the standard for the treatment of STS of the extremities, achieving local control rates of 90% and a 5-year overall survival of 70% [10].

In STS of the extremities, oncological outcomes are similar using neoadjuvant vs. adjuvant radiotherapy (RT); however, neoadjuvant RT is preferred, since this approach significantly reduces chronic complications [13].

Neoadjuvant and adjuvant systemic therapy and isolated limb perfusion (ILP) may be considered in patients at a high risk of metastatic disease or if tumor volume reduction is required to facilitate surgical resection and limb sparing [14].

This study aimed to establish the prognostic factors (PF) associated with tumor recurrence and overall survival (OS) in patients diagnosed with STS of the extremities managed in the Functional Unit for Breast and Soft Tissue Tumors of the INC from January 2008 to December 2018.

## 2. Materials and Methods

An observational, analytical, historical cohort-type study was conducted, which included patients diagnosed with STS of the extremities managed in the Functional Unit for Breast and Soft Tissue Tumors of the INC from 1 January 2008 to 31 December 2018,

who met the following inclusion criteria: over 18 years of age at diagnosis, diagnosis of primary or recurrent extremity STS without distant disease, and surgical management performed by one of the specialists of the Functional Unit during the described time period. To identify the pertinent medical records, a search was made in SIAI of the ICD-10 codes for malignant tumors of the upper and lower extremities, cross-referencing the information with the Functional Unit's surgical scheduling records during the period described to identify the file registration numbers of patients. These registration numbers were then reviewed one by one in the SAP system to identify patients who met the inclusion criteria for the study. Information on sociodemographic and clinicopathological characteristics was taken from the Functional Unit's database and the electronic medical record system. Data were collected by one of the authors and then compiled in an electronic platform designed for the storage of clinical study information (REDCap). The quality and fidelity of the information were evaluated by an assigned supervisor from the Research Division of the INC. The study was approved by the Ethics Committee of the INC (Minute No. IX-023644).

For the descriptive statistical analysis, absolute and relative frequencies, medians, and interquartile ranges (IQR) were estimated for qualitative variables.

As oncological outcomes of interest in the study, overall survival (OS) was evaluated, defined as the time between admission to the Functional Unit and the time of death from any cause. Recurrence-free survival (RFS) was defined as the time between the date of admission to the Functional Unit and the date of diagnosis of local, regional, or systemic recurrence. For statistical analysis, cases of loss or termination of follow-up, without information on the outcomes of interest (recurrence or death), were taken as right censoring. The frequency of acute complications (occurring during the first 30 postoperative days) and chronic complications (occurring after 30 days) were considered as safety outcomes.

Survival functions estimated with the Kaplan–Meier method were used to describe OS and RFS. We evaluated the relationship between the outcomes of interest (OS, RFS, and complications) and a group of variables identified in the literature as possible risk factors for these outcomes (sex, age, clinical stage, type of presentation, histologic type, grade, size and location of the tumor). Cox proportional hazards models were developed to analyze the association between variables and outcomes taken as time to event. Binomial logistic regression models were used for the outcome of chronic complications (yes or no) and Poisson regression models were used for acute complications (number of complications). Hypothesis testing for the statistical models used 5% significance levels. Stata 16$^{®}$ statistical software (serial number 501706364196) was used for statistical analysis.

## 3. Results

During the mentioned study period, 424 patients with tumors of the extremities and trunk were admitted to the Functional Unit of the INC; 127 of them were excluded for having a diagnosis other than STS, as were 70 patients for having a location other than the extremities or due to distant metastatic disease. In the end, 227 patients met the inclusion criteria for the study (Figure 1).

The median age at diagnosis was 53 years (IQR: 18–89); 51.3% (n = 117) of the patients were men. Most of the patients (62.5%, n = 142) were admitted to the Functional Unit without prior treatment, 27.8% (n = 63) with persistent tumors after surgical management at another institution, and 9.7% (n = 22) with recurrence of a pre-existing disease. The predominant anatomical location was the lower limb (77%, n = 175), especially the thigh (47.7%, n = 108); 17.2% (n = 39) of the tumors corresponded to well-differentiated liposarcoma, with the same percentage representing undifferentiated pleomorphic sarcoma. The predominant histologic grade was III in 64.8% (n = 147). Most of the patients (29.2%, n = 66) were in stage IIIB at diagnosis. In relation to tumor size, 74.5% (n = 169) of the patients presented with a size larger than 5 cm at diagnosis. The set of clinicopathological characteristics evaluated in the cohort is summarized in Table 1.

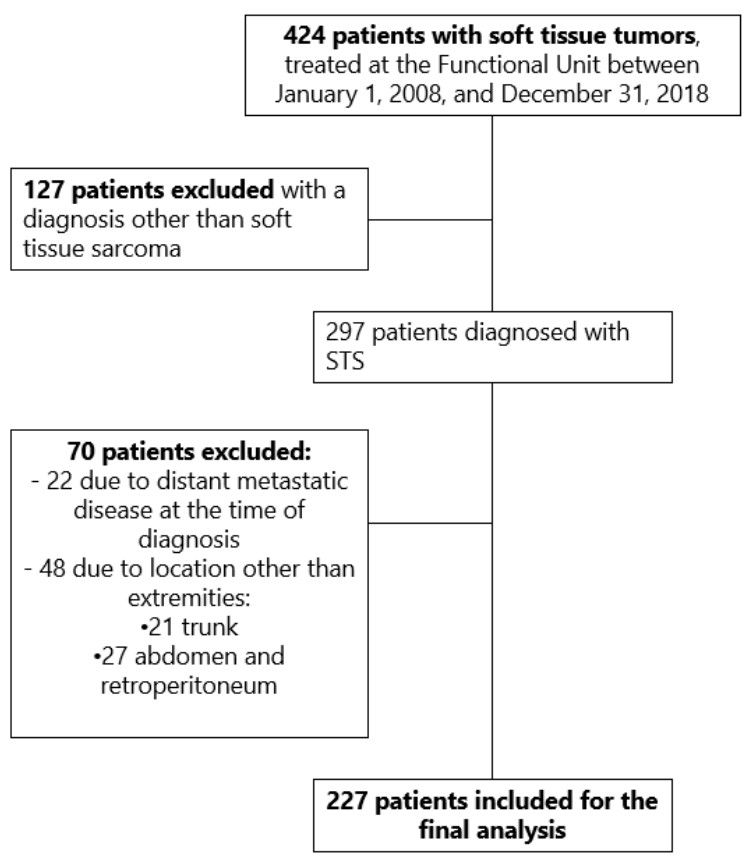

**Figure 1.** Selection of cohort patients.

**Table 1.** Clinicopathological characteristics of cohort patients.

| Characteristics | Total of Patients (n = 227), n (%) |
|---|---|
| **Median age (years)** | 53 (18–89) |
| **Sex** | |
| Men | 117 (51.5) |
| Women | 110 (48.5) |
| **Presentation type** | |
| Primary without treatment | 142 (62.5) |
| Tumor persistence | 63 (27.8) |
| Tumor recurrence | 22 (9.7) |
| **Histologic type** | |
| Well-differentiated liposarcoma | 39 (17.2) |
| Undifferentiated pleomorphic sarcoma | 39 (17.2) |
| Myxoid liposarcoma | 33 (14.5) |
| Synovial sarcoma | 25 (11) |
| Myxofibrosarcoma | 21 (9.3) |
| Leiomyosarcoma | 19 (8.4) |
| Malignant neural sheath tumor | 16 (7) |
| Others | 35 (15.4) |
| **Histologic grade** | |
| I | 67 (29.5) |
| II | 12 (5.3) |
| III | 147 (64.8) |
| No data | 1 (0.4) |

**Table 1.** *Cont.*

| Characteristics | Total of Patients (n = 227), n (%) |
|---|---|
| **Tumor size** | |
| T1 | 53 (23.3) |
| T2 | 64 (28.1) |
| T3 | 40 (17.7) |
| T4 | 65 (28.7) |
| **No data (Initial surgery outside the INC)** | 5 (2.2) |
| **Clinical stage** | |
| IA | 14 (6.2) |
| IB | 43 (18.9) |
| II | 35 (15.4) |
| IIIA | 57 (25.1) |
| IIIB | 66 (29.1) |
| IV (Lymph node involvement) | 3 (1.3) |
| Not applicable (Dermatofibrosarcoma protuberans) | 9 (3.9) |
| **Tumor location** | |
| Thigh | 108 (47.7) |
| Leg | 39 (17.2) |
| Forearm | 23 (10.1) |
| Gluteus | 16 (7) |
| Foot | 13 (5.7) |
| Arm | 12 (5.3) |
| Shoulder | 10 (4.4) |
| Hand | 6 (2.6) |

Regarding the therapeutic strategies used, 74.9% (n = 170) underwent initial surgical treatment, with wide local resection being the most performed procedure (57.4%, n = 130), followed by amputation (23.8%, n = 54). In relation to neoadjuvant treatment, 11.9% of the patients (n = 27) received RT, 10.1% (n = 23) underwent ILP, and 3.1% (n = 7) received neoadjuvant chemotherapy (CHT).

Of the patients, 7% (n = 16) underwent lymph node dissection during the primary tumor surgery due to lymph node chain involvement in the affected limb; 8.3% (n = 19) had sentinel lymph node biopsy. Sentinel lymph node involvement was found in two of these patients, and they underwent lymph node dissection.

There were acute postoperative complications in 26.9% (n = 61) of the patients; the most frequent one was surgical wound dehiscence (39.3%, n = 24). Of these patients, 19.7% (n = 12) had received neoadjuvant RT. The second most frequent complication was infection of the superficial surgical site in 8.4% (n = 19). There were 14 chronic complications (6.1%), the main one being functional limitation of the limb (n = 4), followed by fibrosis (n = 3).

Positive margins were reported in surgical pathology in 14.9% (n = 34) of the patients; six of them underwent resection with planned positive margins and received RT (neoadjuvant in two and adjuvant in four cases). Eleven (4.8%) patients underwent surgery to widen the margins to achieve definitive negative pathology; six of them received neoadjuvant and five received adjuvant RT.

It was not possible to perform margin widening in 12 patients; 5 of them had already received neoadjuvant RT and 7 were referred to adjuvant RT. Only five patients did not accept any other type of treatment and had not received neoadjuvant RT either.

Regarding adjuvant treatment, 37.4% (n = 85) of the patients received adjuvant RT, with doses ranging between 30 and 66 Gy, while intraoperative radiotherapy (IORT) was applied in 6.2% (n = 14), with doses between 12 and 15 Gy. Adjuvant CHT was used in 22.4% (n = 51), with the MAI scheme (mesna, doxorubicin, ifosfamide) being the most

frequently administered combination in 78.2% (n = 43) of the cases. The treatment types are described in Table 2.

**Table 2.** Types of treatment administered to cohort patients.

| Administered Treatment | Total of Patients (n = 227), n (%) |
|---|---|
| **Initial treatment type** | |
| Surgical treatment | 170 (74.9) |
| Neoadjuvant RT | 27 (11.9) |
| ILP | 23 (10.1) |
| Neoadjuvant CHT | 7 (3.1) |
| **Type of primary tumor surgery** | |
| Wide local resection | 130 (57.4) |
| Amputation | 54 (23.8) |
| **Widening of margins of previous non-oncological surgery outside the INC** | 28 (12.3) |
| Compartmental resection | 9 (3.9) |
| Marginal resection | 6 (2.6) |
| **Additional interventions** | |
| Sentinel lymph node | 19 (8.3) |
| Lymph node dissection | 18 (7.9) |
| IORT | 14 (6.1) |
| **Positive margins in INC pathology** | |
| No | 189 (83.3) |
| Yes | 34 (14.9) |
| Planned positive margins | 6 (17.7) |
| **Surgery to achieve negative margins** | |
| Widening of margins | 11 (4.8) |
| **Adjuvant treatment** | |
| Adjuvant CHT | 51 (22.4) |
| Adjuvant RT | 85 (37.3) |

RT: radiotherapy; ILP: isolated limb perfusion; CHT: chemotherapy; IORT: intraoperative radiotherapy.

Regarding the OS analysis, at the time of study closure, 37% (n = 84) of the patients were alive with no evidence of disease, 26% (n = 60) had died from the disease, 18% (n = 40) had died from another cause, 2% (n = 5) remained alive with clinical or imaging evidence of the disease, and 38 (17%) did not complete follow-up.

There was a total of 100 deaths during follow-up, representing a mortality rate of 8.5 deaths per 100 patient-years (CI95%: 7–10.3).

The 227 patients included in the study provided a total of 1179.2 years of follow-up, with a median follow-up of 4.5 years (IQR: 6.1 years). The median OS was 10 years (25th percentile = 2.5 years; 75th percentile not reached) (Figure 2).

The Cox proportional hazards model, performed by combining various clinical and pathological variables related to OS, found a higher risk of mortality at an older age (HR = 1.01; CI95%: 1–1.02; $p$ = 0.048). Histologic grade II (HR = 0.69; CI95%: 0.007–0.67; $p$ = 0.021) and wide local resection (HR = 0.48; CI95%: 0.24 –0.96; $p$ = 0.038) were factors associated with better OS (Table 3).

In relation to RFS, 33% (n = 75) of the cohort patients presented with disease recurrence; 50.6% (n = 38) at the local level. Of these patients, 65.7% (n = 25) underwent new surgical procedures to control the disease, which included wide local resection (n = 13), amputation (n = 11) and compartmental resection (n = 1); 34.2% (n = 13) did not accept additional treatments. When analyzing the group of patients with local recurrence, it was found that nine of them (23.6%) had positive margins, two planned and seven unplanned. Two patients who had received IORT presented with local recurrence; 19.8% (n = 45) had systemic

progression, the most frequently involved organs being lung (80%, n = 36) and bone (13.3%, n = 6). Twenty percent (n = 9) underwent pulmonary metastasectomy and 66.6% (n = 30) received primary CHT.

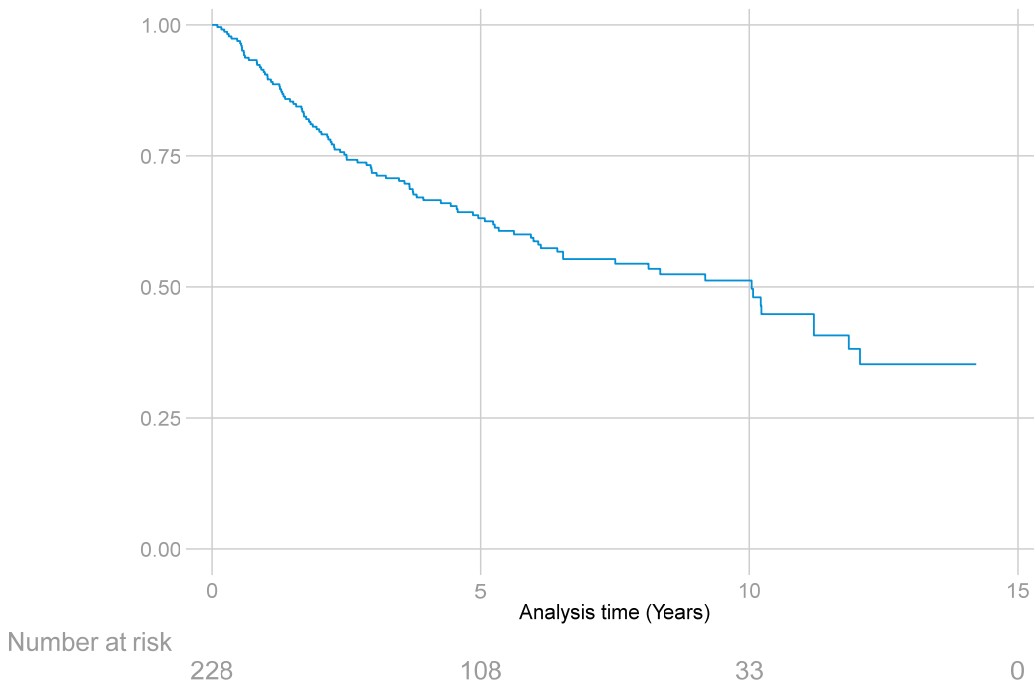

**Figure 2.** Overall survival function estimated with the Kaplan–Meier method in patients with STS of the extremities.

**Table 3.** Adjusted Cox proportional hazards estimates for OS and RFS of all types (local, regional, and distant).

| Variable | Overall Survival Hazard Ratio (IC95%) | | Recurrence-Free Survival Hazard Ratio (IC95%) | |
|---|---|---|---|---|
| **Age** | **1.01 (1–1.02)** | **p = 0.048** | 0.99 (0.983–1.015) | p = 0.92 |
| **Tumor size** | 1.01 (0.99–1.04) | p = 0.18 | 1.03 (0.996–1.056) | p = 0.08 |
| **Histologic grade** | | | | |
| I | Ref. | | Ref. | |
| II | **0.69 (0.007–0.67)** | **p = 0.021** | 1.97 (0.49–7.9) | p = 0.33 |
| III | *0.13 (0.01–1.1)* | p = 0.062 | **5.36 (2.29–12.56)** | **p < 0.001** |
| **Type of primary tumor surgery** | | | | |
| Amputation | Ref. | | Ref. | |
| Compartmental resection | 0.69 (0.2–2.23) | p = 0.56 | 0.68 (0.16–2.83) | p = 0.6 |
| Wide local resection | **0.48 (0.24–0.96)** | **p = 0.038** | 0.56 (0.26–1.21) | p = 0.14 |
| Marginal resection | 1.81 (0.47–6.9) | p = 0.38 | 2.16 (0.41–11.3) | p = 0.36 |
| Widening of margins of previous non-oncological surgery outside the INC | 0.36 (0.12–1.1) | p = 0.075 | **0.16 (0.045–0.6)** | **p = 0.006** |
| **Positive margins in INC pathology** | | | | |
| Yes | 1.11 (0.54–2.26) | p = 0.76 | 1.56 (0.75–3.2) | p = 0.23 |
| No | Ref. | | Ref. | |
| **Presentation type** | | | | |
| Primary without treatment | Ref. | | Ref. | |
| Tumor persistence | 0.95 (0.51–1.77) | p = 0.88 | **2.34 (1.25–4.35)** | **p = 0.007** |
| Tumor recurrence | 0.83 (0.39–1.76) | p = 0.63 | 1.62 (0.74–3.53) | p = 0.21 |

**Table 3.** *Cont.*

| Variable | Overall Survival Hazard Ratio (IC95%) | | Recurrence-Free Survival Hazard Ratio (IC95%) | |
|---|---|---|---|---|
| **Initial treatment type** | | | | |
| Surgical treatment | Ref. | | Ref. | |
| Neoadjuvant CHT | 1.18 (0.43–3.24) | *p* = 0.73 | 0.68 (0.15–3.03) | *p* = 0.61 |
| Neoadjuvant RT | 0.7 (0.3–1.65) | *p* = 0.42 | 0.86 (0.35–2.09) | *p* = 0.75 |
| ILP | 0.75 (0.36–1.54) | *p* = 0.44 | 0.62 (0.25–1.52) | *p* = 0.75 |
| **Adjuvant treatment** | | | | |
| Adjuvant CHT | | | | |
| Yes | 0.66 (0.38–1.14) | *p* = 0.14 | 1.14 (0.3–1.2) | *p* = 0.15 |
| No | *Ref.* | | *Ref.* | |
| Adjuvant RT | | | | |
| Yes | 0.7 (0.38–1.27) | *p* = 0.24 | 0.6 (0.3–1.2) | *p* = 1.2 |
| No | *Ref.* | | *Ref.* | |

Ref: reference; CHT: chemotherapy; RT: radiotherapy; ILP: isolated limb perfusion.

For RFS of any type (local, regional, or distant), the 227 patients provided a total of 945 years of follow-up. The median follow-up was 2.9 years (IQR: 0.1–14.2). The median survival could not be estimated because less than 50% of the patients had this outcome (Figure 3).

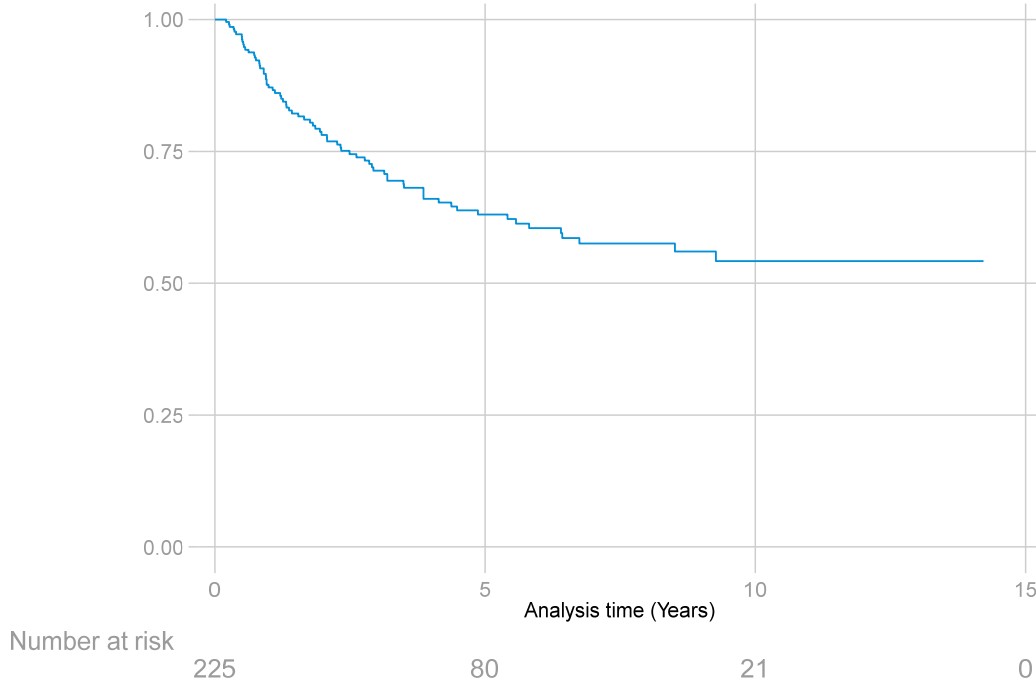

Number at risk
| 225 | 80 | 21 | 0 |

**Figure 3.** Recurrence-free survival function estimated with the Kaplan–Meier method in patients with STS of the extremities.

The median RFS of any type, as well as the specific median for local, regional, or distant recurrence, could not be estimated, since less than 50% of the patients presented this type of outcome.

The Cox proportional hazards model for RFS of any type (local, regional, or distant) showed that tumor persistence at the time of admission to the INC (HR = 2.34; CI95%: 1.25–4.35; *p* = 0.007) and histologic grade III (HR = 5.36; CI95%: 2.29–12.56; *p* < 0.001) were factors associated with higher disease recurrence. By contrast, surgery to widen the margins of previous non-oncological surgery performed outside the INC (HR = 0.16; CI95%:

0.045–0.6; *p* = 0.006) was considered a protective factor. Neither neoadjuvant nor adjuvant CHT or RT treatment had an impact on the oncological outcomes measured in this cohort (Table 3).

When analyzing the variables related to local recurrence in the Cox proportional hazards model, it was found that tumor persistence at the time of admission to the INC (HR = 2.85; CI95%: 1.23–6.57; *p* = 0.014) and histologic grade III (HR = 6.09; CI95%: 2.03–18.2; *p* = 0.001) were factors associated with higher local recurrence.

Regarding distant recurrence, an unfavorable association was found with tumor size (HR = 1.03; CI95%: 1.007–1.067; *p* = 0.015) and histologic grade III (HR = 4.53; CI95%: 1.42–14.49; *p* = 0.01). Wide local resection (HR = 0.37; CI95%: 0.14–0.94; *p* = 0.03) and the widening of margins of previous extra-institutional surgery (HR = 0.07; CI95%: 0.008–0.64; *p* = 0.019) behaved as protective factors of distant recurrence (Table 4).

**Table 4.** Adjusted Cox proportional hazards estimates for local and distant recurrence.

| Variable | Local Recurrence Hazard Ratio (IC95%) | | Distant Recurrence Hazard Ratio (IC95%) | |
|---|---|---|---|---|
| **Age** | **0.99 (0.77–1.02)** | *p* = 0.77 | 0.99 (0.983–1.019) | *p* = 0.93 |
| **Tumor size** | 1 (0.95–1.05) | *p* = 0.89 | **1.03 (1.007–1.067)** | **p = 0.015** |
| **Histologic grade** | | | | |
| I | Ref. | | Ref. | |
| II | 1.1 (0.11–10.5) | *p* = 0.92 | 2.21 (0.37–13.08) | *p* = 0.38 |
| III | *6.09 (2.03–18.2)* | *p = 0.001* | **4.53 (1.42–14.49)** | **p = 0.011** |
| **Type of primary tumor surgery** | | | | |
| Amputation | Ref. | | Ref. | |
| Compartmental resection | 3.43 (0.48–24.63) | *p* = 0.21 | 0.48 (0.086–2.74) | *p* = 0.41 |
| Wide local resection | 1.69 (0.46–6.1) | *p* = 0.42 | **0.37 (0.14–0.94)** | **p = 0.038** |
| Marginal resection | 8.88 (0.75–104.4) | *p* = 0.08 | 0.78 (0.08–7.52) | *p* = 0.83 |
| Widening of margins of previous non-oncological surgery outside the INC | 0.51 (0.08–3.06) | *p* = 0.46 | **0.07 (0.008–0.64)** | ***p = 0.019*** |
| **Positive margins in INC pathology** | | | | |
| Yes | 1.74 (0.69–4.34) | | 1 (0.37–2.72) | |
| No | Ref. | *p* = 0.23 | Ref. | *p* = 0.98 |
| **Presentation type** | | | | |
| Primary without treatment | Ref. | | Ref. | |
| Tumor persistence | **2.85 (1.23–6.57)** | **p = 0.014** | 1.49 (0.65–3.41) | *p* = 0.33 |
| Tumor recurrence | 1.6 (0.5–5.07) | *p* = 0.42 | 1.42 (0.55–3.64) | *p* = 0.46 |
| **Initial treatment type** | | | | |
| Surgical treatment | Ref. | | Ref. | |
| Neoadjuvant CHT | 0.81 (0.09–6.84) | *p* = 0.85 | 1.35 (0.28–6.44) | *p* = 0.7 |
| Neoadjuvant RT | 0.77 (0.21–2.65) | *p* = 0.65 | 0.98 (0.32–3.03) | *p* = 0.98 |
| ILP | 1.02 (0.32–3.24) | *p* = 0.96 | 0.50 (0.14–1.78) | *p* = 0.29 |
| **Adjuvant treatment** | | | | |
| Adjuvant CHT | | | | |
| Yes | 0.75 (0.33–1.69) | *p* = 0.49 | 1.25 (0.62–2.54) | *p* = 0.52 |
| No | Ref. | | Ref. | |
| Adjuvant RT | | | | |
| Yes | 0.55 (0.22–1.4) | *p* = 0.21 | 0.82 (0.33–2.01) | *p* = 0.66 |
| No | Ref. | | Ref. | |

Ref.: reference; CHT: chemotherapy; RT: radiotherapy; ILP: isolated limb perfusion.

It was not possible to develop the Cox logistic regression model to estimate the variables related to regional recurrence, since few of these events occurred in this cohort.

*Acute and Chronic Complications*

According to the Poisson regression model for acute complications, widening the margins of extra-institutional surgery increases the number of acute complications by 69% ($\beta$ = 1.69; CI95%: 0.34–3.05; $p$ = 0.014), wide local resection increases it by 38% ($\beta$ = 1.38; CI95%: 0.43–2.33; $p$ = 0.004), compartmental resection by 37% ($\beta$ = 1.37; CI95%: 0.1–2.63; $p$= 0.03), and tumor relapse by 9.5%, ($\beta$ = 0.95; CI95%: 0.32–1.59; $p$ = 0.003). The number of acute complications also increases with older age ($\beta$ = 0.017; CI95%: 0.002–0.031; $p$ = 0.022) and larger tumor size ($\beta$ = 0.036; CI95%: 0.007–0.065; $p$= 0.015) (Table 5).

**Table 5.** Poisson regression model for acute complications and logistic regression model for chronic complications.

| Variable | Acute Complications Coefficient ($\beta$) (IC95%) | | Chronic Complications Odds Ratio (IC95%) | |
|---|---|---|---|---|
| **Age** | **0.017 (0.002–0.031)** | ***p* = 0.022** | 1 (0.96–1.03) | *p* = 0.89 |
| **Tumor size** | **0.036 (0.007–0.065)** | ***p* = 0.015** | 0.96 (0.87–1.07) | *p* = 0.52 |
| **Histologic grade** | | | | |
| I | Ref. | | Ref. | |
| II | 0.19 (0.82–1.22) | *p* = 0.7 | 1.1 (0.07–15.76) | *p* = 0.94 |
| III | 0.41 (0.19–1.02) | *p* = 0.18 | 1.55 (0.3–7.9) | *p* = 0.59 |
| **Type of primary tumor surgery** | | | | |
| Amputation | Ref. | - | Ref. | |
| Compartmental resection | **1.37 (0.1–2.63)** | ***p* = 0.03** | - | |
| Wide local resection | **1.38 (0.43–2.33)** | ***p* = 0.004** | 2.49 (0.23–26.46) | *p* = 0.44 |
| Marginal resection | 1.47 (−0.12–3.06) | *p* = 0.07 | 7.33 (0.24–219.49) | *p* = 0.25 |
| Widening of margins of previous non-oncological surgery outside the INC | **1.69 (0.34–3.05)** | ***p* = 0.014** | 0.77 (0.027–21.82) | *p* = 0.88 |
| **Positive margins in INC pathology** | | | | |
| Yes | Ref. | | *Ref.* | |
| No | −0.07 (−0.754–0.596) | *p* = 0.81 | 0.62 (0.11–3.37) | *p* = 0.58 |
| **Presentation type** | | | | |
| Primary without treatment | Ref. | | Ref. | |
| Tumor persistence | −0.24 (−1.01–0.52) | *p* = 0.53 | 1.53 (0.32–7.25) | *p* = 0.59 |
| Tumor recurrence | **0.95 (0.32–1.59)** | ***p* = 0.003** | 3.35 (0.68–16.33) | *p* = 0.13 |
| **Initial treatment type** | | | | |
| Surgical treatment | Ref. | | Ref. | |
| Neoadjuvant CHT | −0.14 (−1.63–1.34) | *p* = 0.84 | 2.31 (0.18–29.12) | *p* = 0.51 |
| Neoadjuvant RT | 0.16 (−0.05–0.88) | *p* = 0.65 | 3.53 (0.44–28.42) | *p* = 0.23 |
| ILP | −0.21 (−1.18–0.76) | *p* = 0.67 | 2.41 (0.37–15.68) | *p* = 0.35 |
| **Adjuvant treatment** | | | | |
| Adjuvant CHT | | | | *p* = 0.45 |
| Yes | 0.02 (−0.57–0.62) | *p* = 0.93 | 0.55 (0.11–2.64) | |
| No | Ref. | | Ref. | |
| Adjuvant RT | | | | ***p* = 0.052** |
| Yes | −0.03 (−0.6–0.53) | *p* = 0.9 | **4.36 (0.98–19.26)** | |
| No | Ref. | | Ref. | |

Ref: reference; CHT: chemotherapy; RT: radiotherapy; ILP: isolated limb perfusion.

In the logistic regression model, the odds ratio (OR) of chronic complications in patients who received adjuvant RT was 4.36 (CI95%: 0.98–19.26; $p$ = 0.052). No relationship was found between this type of complication and the histopathological characteristics or treatment received (Table 5).

## 4. Discussion

To date, this is the first study that examines treatment experience results in patients with STS of the extremities managed in a Colombian reference cancer center, where most of the patients are admitted with advanced disease.

Surgical treatment is the mainstay of treatment for patients with non-metastatic STS and for those with resectable metastases, independent of histologic subtype and location, with its main objectives being local control of the disease by achieving adequate surgical margins and the preservation of limb function [2].

In STS, there are several clinical and histopathological factors associated with adverse oncological outcomes, which have allowed predicting the clinical course of the disease.

In this cohort, presentation type with persistent disease and high histologic grade were identified as the most important prognostic factors for RFS.

Patients admitted to the INC with tumor persistence after previous extra-institutional non-oncologic surgery had a higher risk of recurrence of any type and a higher risk of local relapse compared to patients who presented with primary disease, which relates to what has been described in series such as Blay et al. [15], who report lower rates of local recurrence, disease progression, and death in patients initially treated in high-volume centers by surgeons specialized in the management of STS.

Approximately 25% of patients with STS of the extremities develop distant metastases after surgical resection with negative margins [16]. This incidence increases to 50% when high-risk factors are combined, such as tumor size > 5 cm, deep fascia tumors, and intermediate or high histologic grade [5,17]. In this cohort, most patients (64.8%) had histologic grade III, which, in turn, was more related to worse RFS of any type and higher local and distant recurrence. This finding is similar to those described in series such as Coindre et al. [18], Brennan et al. [19], Torosian et al. [20] and Ruo-He Li et al. [21], where a high histologic grade was related to worse distant metastasis-free survival rates, without being associated with worse OS rates.

The relationship between tumor size and local relapse has been controversial. In this cohort, 74.3% of the patients presented with a tumor size larger than 5 cm, and no statistically significant association was found with local relapse and OS, but a statistical association was found with distant recurrence. Several series have found that tumor size does not have a significant influence on local control, but it does have a relationship with distant recurrence and disease-specific survival [22–25].

According to the World Health Organization (WHO) classification, there are more than 100 different histologic subtypes of STS, each of which has different clinical and prognostic characteristics. Any histologic subtype can develop in the extremities; in this cohort, the most frequent types were well-differentiated liposarcoma, undifferentiated pleomorphic sarcoma, and myxoid liposarcoma. Given the heterogeneity and multiple histologic subtypes described, it was not possible to include this variable in the multivariate analysis. Thus, it was difficult to establish a direct relationship between each of these subtypes and the different oncological outcomes described.

Microscopically positive margins are known to be associated with a higher rate of local recurrence. In this cohort, with a local recurrence rate of 16.7%, a weak association was found with marginal resection, but no statistically significant association was shown between positive surgical margins and local or distant recurrence. This relates to what is described in studies such as Gronchi et al. [26], where surgical margins had no impact on local and distant recurrence ($p = 0.179$) in the first years of follow-up. However, they were associated with worse local recurrence rates after 5 years of follow-up, similar to that reported by Stojadinovic et al. [27], where an impact on local and distant recurrence was only found 2 years after resection of the primary tumor. In this cohort, 14.9% of the patients had positive surgical margins. It is likely that the lack of statistical significance with adverse oncological outcomes is due to some additional surgical intervention to achieve negative oncological margins. In addition, 2.6% of these patients had planned positive margins due

to planned marginal resection because of neurovascular compromise, and all these patients received neoadjuvant or adjuvant RT.

Regarding the type of surgical procedure, an association close to statistical significance was found with marginal resection and local recurrence, but no differences were found between procedure type and the oncological outcomes analyzed. Most studies worldwide have failed to establish a relationship between the type of surgical procedure and survival [27]. Series like Trovik et al. [28] and Zagars et al. [29] report intrinsic tumor characteristics, such as size, depth, histologic grade, and histologic subtype, as factors related to distant recurrence and disease-specific survival, rather than the type of procedure performed.

In this cohort, lymph node involvement was present in 7.9% of the patients who underwent lymph node dissection during primary tumor surgery due to lymph node chain involvement in the affected limb or after a positive sentinel lymph node biopsy. Nodal involvement in STS is rare, with reported rates of 2 to 10% in all histopathological subtypes [30], which relates to what was found in the present study.

Although the role of radiation is well established in STS of the extremities, the optimal sequence of radiation surgery in terms of oncological outcomes has not yet been defined [31]. In the study by O'Sullivan et al. [32] comparing preoperative and postoperative RT, neoadjuvant treatment revealed a slightly significant improvement in OS compared to postoperative treatment. In this cohort, no relationship was found between the time of administration of RT and OS, although a relationship was found between adjuvant RT and an increased risk of chronic complications, which relates to what has been described in the study by Davis et al. [32], where postoperative RT was associated with a higher incidence of fibrosis, joint stiffness, and reduced limb functionality [32,33]. In this cohort, 37.4% (n = 85) of the patients received adjuvant RT and intraoperative radiotherapy (IORT) was applied in 6.2% (n = 14). Adjuvant CHT was used in 22.4% (n = 51), while neoadjuvant RT was applied in 11.9%, due to the study period (2008–2018). At present, the Multidisciplinary Board of Sarcomas has a greater use of radiotherapy in the neoadjuvant setting; thus, in the Functional Unit, patients with sarcomas of the extremities, especially well-differentiated liposarcomas with borderline resectability and grade 2 and 3 dedifferentiated tumors, are referred to neoadjuvant radiotherapy and subsequent surgical management for limb salvage. In addition, we use the Sarculator nomogram to define the benefit of neoadjuvant chemotherapy in selected cases in a multidisciplinary meeting.

Isolated limb perfusion allowed performing limb-sparing surgery in 47.8% of the patients who underwent this type of intervention, being a therapeutic alternative in patients with STS whose tumors make conservative surgery difficult due to their extension (multifocal or multicompartmental) or volume.

This study showed that the only variable associated with higher mortality was age; OS could be affected by age not only in relation to the clinical course of the disease, but also by unrelated concurrent morbidity.

As for acute complications, they occurred in 26.9% of patients in this cohort, which is similar to what is described in the literature, with rates ranging from 11 to 29% [33]. It was found that the widening of margins from previous extra-institutional surgery, wide local resection, and compartmental resection increase the risk of a greater number of acute complications, as well as tumor recurrence, which has been previously reported in series such as Schwartz et al. [34], where prolonged surgical time was related to a higher risk of infection and surgical wound dehiscence.

## 5. Study Limitations

One of the main limitations of this study is its retrospective nature; additionally, in some cases, the follow-up was short.

Due to the heterogeneity of the histologic subtypes, it was not possible to include this variable in the multivariate analysis; thus, it was not possible to establish a relationship between this variable and the outcomes evaluated.

## 6. Conclusions

Soft tissue sarcomas are low-incidence tumors, comprising a heterogeneous group of neoplasms with diverse outcomes determined by several factors. In this cohort, tumor persistence was a determining prognostic factor, so these types of tumors should ideally be managed in referral centers by an experienced multidisciplinary team seeking to improve oncological outcomes.

**Author Contributions:** Conceptualization, S.E.D.C. and J.M.V.; methodology, R.S.P.; software, data manager from the Research Division of the INC; validation, J.M.V., S.E.D.C. and supervisor from the Research Division of the INC; formal analysis, R.S.P., S.E.D.C. and J.M.V.; investigation, S.E.D.C., J.M.V., C.L.M., I.M.L., M.G.M., J.Á.A., R.S.R. and C.A.D.T.; resources, INC; data curation, S.E.D.C., J.M.V. and R.S.P.; writing—original draft preparation, S.E.D.C., J.M.V. and R.S.P.; writing—review and editing, S.E.D.C., J.M.V., C.L.M. and I.M.L.; visualization, Research Division of the INC; supervision, Research Division of the INC; project administration, Research Division of the INC.; funding acquisition, Research Division of the INC. All authors have read and agreed to the published version of the manuscript.

**Funding:** This research was funded by the National Cancer Institute Bogota/Colombia.

**Institutional Review Board Statement:** This study was conducted in accordance with the Declaration of Helsinki, and approved by the Ethics Committee of the Instituto Nacional de Cancerologia/Bogota/Colombia, Minute No. IX-023644. Date of approval: 7 July 2021. Initiation document: 24 August 2021.

**Informed Consent Statement:** This study does not have informed consent because it is a retrospective cohort. All patients provided informed consent for surgical treatment.

**Data Availability Statement:** Study information on sociodemographic and clinicopathological characteristics was taken from the Functional Unit's database and the electronic medical record system. Data were collected by one of the authors and then compiled in an electronic platform designed for the storage of clinical study information (REDCap). The quality and fidelity of the information were evaluated by an assigned supervisor from the Research Division of the INC.

**Acknowledgments:** Data manager from the Research Division of the INC and Supervisor from the Research Division of the INC.

**Conflicts of Interest:** The authors declare no conflict of interest.

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
