# Peer review of "Prognostic Factors Associated with Tumor Recurrence and Overall Survival in Soft Tissue Sarcomas of the Extremities in a Colombian Reference Cancer Center"

_curroncol, doi:10.3390/curroncol31040131_

Round 1
Reviewer 1 Report
Comments and Suggestions for Authors
This study conducted by Sandra E. Díaz Casas et. al. at the INC in Colombia, concentrated on soft tissue sarcomas in the extremities, which are low-incidence tumors with factors linked to adverse oncological outcomes. The research sought to assess prognostic factors linked to tumor recurrence and overall survival in STS 227 patients treated at INC from January 2008 to December 2018. The analysis identified age, tumor persistence upon admission, and histologic grade III as noteworthy factors in recurrence. Furthermore, both tumor size and histologic grade III were correlated with an elevated risk of distant recurrence.
The manuscript is well written and easy to follow, but the reviewer would suggest the authors mention how different these results are compared to similar studies conducted on different cohorts like those mentioned in the reference like 19 (1986;58(2):306-9) and 20 (260(3), 416).
The introduction, methodology, result and discussion section are detailed and the manuscript is fit for publication.
Author Response
Thanks for your evaluation. The histological grade is a relevant factor for staging and is related with disease free survival and overal survival. In this cohort most of patients had histological grade III. That is what we mean.
Thank you for your comments. We have included two more references to relate the prognostic factors to the cohort results.
Reviewer 2 Report
Comments and Suggestions for Authors
The paper Prognostic factors associated with tumor recurrence and OS in STS of the extremities reproduces the classical retrospective research resuming the activity of a Center devoted to STS care.
The introduction is complete and clear.
Material and methods session is not satisfactory: 424 patients treated in more than 10 years of activity means about 40 patients/ year. This number is very far from the optimal number of patients per year as stated by Blay and coll (Blay JY, Honoré C, Stoeckle E, Meeus P, Jafari M, Gouin F, et al. Surgery in reference centers improves survival of sarcoma patients: a nationwide study. Ann Oncol. 2019;30(7):1143-53) in which the mean number is around 100 case per year or more.
Moreover , only 227 patients are included in the final analysis , very far from the optimal level.
As consequence of this weakness the results can be considered as preliminary and not definitive .
No relationship are reported between the primary treatment ( surgery upfront, neoadjuvant CT followed by surgery,adjuvant CT , Adjuvant RT ) and overall survival and PFS. We cannot answer to the question: which is the best therapeutic strategy in STS of the extremities?
In some Patients a nodal dissection was performed. It is unusual in STS treatment . Which was the advantages for the Patients? What about complication? Is this technique extensible or must be restricted only to positive adenopaties?
From the analysis of the biggest international groups treating STS we know that OS and LR are related with volume of the tumor, anatomical position, grading, status of the margins, histology, radical intervention. Adjuvant and neoadjuvant therapy are less important ( see ESMO Guidelines 2021)
Probably as consequence of the small number of cases only tumor size, grading and radicality of the intervention are significant in a univariate analysis. Multivariate analysis because of rhe number of samples is not possible,
The conclusions are correct but are not completely supported by the data of this study.
In my opinion a different title as " Ten years of activity on STS of the extremities. A national cancer Institute experience in Colombia" could improve the presentation
Author Response
Thank you for your comments.
Material and methods session is not satisfactory: 424 patients treated in more than 10 years of activity means about 40 patients/ year. This number is very far from the optimal number of patients per year as stated by Blay and coll (Blay JY, Honoré C, Stoeckle E, Meeus P, Jafari M, Gouin F, et al. Surgery in reference centers improves survival of sarcoma patients: a nationwide study. Ann Oncol. 2019;30(7):1143-53) in which the mean number is around 100 case per year or more.
Moreover, only 227 patients are included in the final analysis, very far from the optimal level.
As consequence of this weakness the results can be considered as preliminary and not definitive.
Colombia is a small country with 50 million inhabitants. Sarcomas are low incidence tumors (less than 1% of all cancers). The Instituto Nacional de Cancerología is a cancer center, and the Functional Unit only treats sarcomas of the extremities, trunk, and retroperitoneum. The orthopedic oncology service treats patients with bone sarcomas. We currently receive an average of 35 new cases of patients with sarcomas of the extremities and 25 to 30 with retroperitoneal sarcomas, which are not negligible numbers for this type of pathology. In this 10-year cohort, of the 425 records from the initial search of the SAP medical records system, 127 patients were excluded at entry because they had bone sarcomas, fibromatosis, skin tumors, and benign soft tissue tumors. The other 70 patients who were excluded were admitted with metastatic disease or sarcomas of the trunk or retroperitoneum. In the end, 227 patients met the inclusion criteria of the study.
No relationship are reported between the primary treatment (surgery upfront, neoadjuvant CT followed by surgery, adjuvant CT, Adjuvant RT) and overall survival and PFS. We cannot answer to the question: which is the best therapeutic strategy in STS of the extremities?
The paragraph before Table 3 includes this description: Neither neoadjuvant nor adjuvant CHT or RT treatment had an impact on the oncological outcomes measured in this cohort (Table 3).
In this cohort, neither radiotherapy nor neoadjuvant or adjuvant chemotherapy had an impact on the oncological outcomes of the patients, that is, neither on disease-free nor on overall survival.
The best therapeutic strategy for the treatment of extremity sarcomas is surgery. In this cohort, we have highlighted that most patients are in advanced clinical stages, with large tumors. For the study period (2008-2018), the evidence at that time indicated the use of adjuvant radiotherapy for high-grade tumors larger than 5 cm or with positive or close borders, whereas for chemotherapy the benefit was very marginal in disease free survival but not in overall survival in high histologic grade tumors. Thus, in this study, only 11.9% of the patients received neoadjuvant radiotherapy, while 37.3% received it as adjuvant treatment. Chemotherapy was administered in only 3.1% of patients in neoadjuvant setting and in 22.4% as adjuvant treatment. At present, the multidisciplinary board of sarcomas has a greater use of radiotherapy in the neoadjuvant setting, thus in the Functional Unit, patients with sarcomas of the extremities, especially well-differentiated liposarcomas with borderline resectability and grade 2 and 3 dedifferentiated tumors, are referred to neoadjuvant radiotherapy and subsequent surgical management for limb salvage. In addition, we use the Sarculator nomogram to define the benefit of neoadjuvant chemotherapy in selected cases in a multidisciplinary meeting. This is also described in the article.
In some Patients a nodal dissection was performed. It is unusual in STS treatment, which was the advantages for the Patients? What about complication? Is this technique extensible or must be restricted only to positive adenopaties?
Sarcomas have hematogenous dissemination, but there are some histological subtypes, such as rhabdomyosarcoma, clear cell sarcoma, epithelioid sarcoma, synovial sarcoma, and angiosarcoma, that lead to lymph node metastasis. In the Functional Unit, epithelioid sarcomas and angiosarcomas with negative lymph node involvement undergo sentinel lymph node biopsy; this procedure was performed in 8.3% of the patients in the cohort. Lymphadenectomy was performed in 7.9% of patients with histologically proven axillary or inguinal metastatic involvement. The Functional Unit has very good expertise in lymphadenectomy because, in addition to sarcomas of the extremities, trunk and retroperitoneum, we also handle patients with breast cancer and melanoma, so morbidity is very low.
From the analysis of the biggest international groups treating STS we know that OS and LR are related with volume of the tumor, anatomical position, grading, status of the margins, histology, radical intervention. Adjuvant and neoadjuvant therapy are less important (see ESMO Guidelines 2021)
Probably as consequence of the small number of cases only tumor size, grading and radicality of the intervention are significant in a univariate analysis. Multivariate analysis because of the number of samples is not possible,
Indeed, in this cohort, histologic grade, tumor size, and advanced age were related to disease recurrence, but we highlight a little-considered factor, which is the type of disease presentation. The Functional Unit receives patients from all over the country, who are people of a low socioeconomic level, which is why they are admitted with surgeries performed in non-oncological hospitals and with advanced disease. In this cohort, 37.5% of the patients had disease persistence at admission after a first procedure performed in another institution, or tumor recurrence. Similarly, this group of patients had a higher disease recurrence. On the other hand, neither chemotherapy nor radiotherapy in the neoadjuvant or adjuvant scenarios had an impact on the oncologic outcomes of the patients in this cohort, which is related to the evidence shown in the guidelines. Cox proportional hazards models were developed to analyze the association between variables and outcomes taken as time to event
The conclusions are correct but are not completely supported by the data of this study.
With all due respect, I believe the overall conclusion is clear and shows the importance of managing this pathology in high-volume cancer centers. This work has demonstrated that in addition to already established prognostic factors such as histologic grade, tumor size, and patient age, the initial surgical approach by a multidisciplinary oncology group with adequate surgical experience and training impacts the prognosis of patients with sarcomas of the extremities.
In my opinion a different title as " Ten years of activity on STS of the extremities. A national cancer Institute experience in Colombia" could improve the presentation
All research protocols have been approved by the Ethics Committee, and the research was presented with this title.
Reviewer 3 Report
Comments and Suggestions for Authors
The manuscript is clear and presented in a well-structured manner.
Because this type of pathology is not a frequent one, a retrospective analysis coming from a big institution focused on this pathology and with national coverage is a welcome event in the publishing area.
The studies have to be continued and extended to larger series.
Author Response
Thank you very much for your comments. I appreciate it very much because preparing these research protocols in my hospital is very difficult.
Reviewer 4 Report
Comments and Suggestions for Authors
The authors of this manuscript performed a retrospective study on a cohort of 227 patients with soft tissue sarcomas of the extremities in order to highlight prognostic factors associated with tumor recurrence and overall survival. They reported that both histologic grade and tumor size were correlated with recurrence risk, as well as tumor persistence after previous extra-institutional non-oncologic surgery. Regarding OS, the only variable associated with higher mortality was age.
Despite the reported findings do not hold high novelty, this study is well perfomed, data are clearly presented and conclusions are supported by results. The manuscript is well written and organised, English is fine. Study limitations are highlighted.
As a suggestion, the authors could have integrated their analysis including inflammatory indices. Indeed, the systemic inflammatory status, represented by circulating immune cells such as NLR, PLR, LMR, is considered a prognostic index in various neoplasms including sarcomas. Some studies showed a worse prognosis in terms of overall suvival, disease-specific survival, and disease-free survival in patients with elevated NLR values. A high PLR also demonstrated a worse prognosis for OS and DFS, while patients with a low LMR showed worse OS and DFS. In addition, LMR also showed to have a predictive role on chemotherapy efficacy (10.1002/ijc.28677, doi.org/10.1038/s41598-018-30442-5, 10.18632/oncotarget.3283, doi.org/10.3390/cancers15041080). If the data about this indexes are not available for these patients, authors should at least discuss this topic as future investigation.
Author Response
As a suggestion, the authors could have integrated their analysis including inflammatory indices. Indeed, the systemic inflammatory status, represented by circulating immune cells such as NLR, PLR, LMR, is considered a prognostic index in various neoplasms including sarcomas. Some studies showed a worse prognosis in terms of overall survival, disease-specific survival, and disease-free survival in patients with elevated NLR values. A high PLR also demonstrated a worse prognosis for OS and DFS, while patients with a low LMR showed worse OS and DFS. In addition, LMR also showed to have a predictive role on chemotherapy efficacy (10.1002/ijc.28677, doi.org/10.1038/s41598-018-30442-5, 10.18632/oncotarget.3283, doi.org/10.3390/cancers15041080). If the data about this indexes are not available for these patients, authors should at least discuss this topic as future investigation.
Thank you very much for your comments; I value them very much. With respect to your suggestion, unfortunately, the Instituto Nacional de Cancerología does not have this technology in the pathology laboratory.
Reviewer 5 Report
Comments and Suggestions for Authors
This is an interesting and important study on STS treatment and outcomes in a single tertiary center in Colombia. I believe this will be of interest to the general audience, however, the manuscript could be strengthened after integrating the following recommendations:
Can shorten introduction.
Methods:
- detail type of data collected
- How are complications graded/recorded/evaluated?
- What data was included in multivariate analyses, what was the criteria for inclusion in the multivariate models?
Results:
- So how do you explain the difference in number of patients between July 2020-2022: 125STS new cases but between 2008-2018 297 new cases?
- Population is under-treated, why did so few patients receive neoadjuvant chemotherapy and radiation therapy?
- High rate of lymph node involvement, how do you explain this?
- Long term complications are not graded?
- In OS analysis paragraph: why does the total % not equal 100%? (Page 7 paragraph 1)
- So there is a discrepancy between text and figure 3: only 33% of patients relapsing, but on the KM curve, it looks like it’s almost 50%?
- You need tables with number at risk beneath your KM curves
- It is overall hard to understand when the authors are referring to univariate versus multivariate analyses. This should be explained/stated a little better in the results section.
Comments on the Quality of English Language
The manuscript is well written but I think will benefit from some language editing to improve the ease of reading.
Author Response
Can shorten introduction.
Thank you for your comments. Regarding the introduction, as it is a global summary of the prognostic factors and treatment of extremity sarcomas, I think it is adequate.
Methods:
- detail type of data collected
The Instituto Nacional de Cancerología (INC) is the principal cancer center in Colombia and a reference center for the whole country. It has a strong research group and a medical records system called SAP where all the clinical information of patients is stored: medical records of each of the Functional Units (breast and soft tissue tumors, clinical oncology, radiotherapy, etc.), surgical descriptions, radiotherapy records, chemotherapy records, anatomy-pathology reports, emergency room admission records, diagnostic images, laboratory results, etc. In addition, all diagnoses are coded by ICD-10. To identify the pertinent medical records, a search was made in SIAI of the ICD-10 codes for malignant tumors of the upper and lower extremities, cross-referencing the information with the Functional Unit’s surgical scheduling records during the period described to identify the file registration numbers of patients. These registration numbers were then reviewed one by one in the SAP system to identify patients who met the inclusion criteria for the study. Subsequently, a database was designed with the study variables in the REDCap platform. All information in the database was verified by the INC’s monitoring group.
- How are complications graded/recorded/evaluated?
All patients taken to surgery have a record of their evolution in the clinical history in SAP system, and all early and late complications are recorded in it. In addition, the INC has an infection committee and a quality group that follows up on infectious complications.
- What data was included in multivariate analyses, what was the criteria for inclusion in the multivariate models?
Survival functions estimated with the Kaplan-Meier method were used to describe OS and RFS. We evaluated the relationship between the outcomes of interest (OS, RFS and complications) and a group of variables identified in the literature as possible risk factors for these outcomes (Sex, age, clinical stage, type of presentation, histologic type, grade, size and location of the tumor). Cox proportional hazards models were developed to analyze the association between variables and outcomes taken as time to event. Binomial logistic regression models were used for the outcome of chronic complications (yes or no), and Poisson regression models were used for acute complications (number of complications). Hypothesis testing for the statistical models used 5% significance levels. Stata 16® statistical software was used for statistical analysis.
Results:
- So how do you explain the difference in number of patients between July 2020-2022: 125STS new cases but between 2008-2018 297 new cases?
I will describe briefly the context of the operation of our Functional Unit. Before 2013, it was a service of oncological surgeons who treated breast cancer, melanoma, and sarcomas. Nevertheless, we managed to better organize the entire INC in Functional Units by pathology, so we created in 2013 the Functional Unit of Breast Tumors and established a database where all patients admitted for the first time with a diagnosis of breast cancer are registered and have a rigorous follow-up by a nurse administrator. By 2020, the Functional Unit of Sarcomas was organized; since that year, we have all the information of patients admitted for the first time with a diagnosis of sarcomas of the extremities and retroperitoneum. This is why, in 2 years, we had the certainty of the number of new cases: of the 125 sarcomas, 62 were of the extremities, and the others corresponded to sarcomas of the trunk and retroperitoneum. In this 10-year study cohort, of the 425 records from the initial search of the SAP medical records system, 127 patients were excluded at entry because they had bone sarcomas, fibromatosis, skin tumors, and benign soft tissue tumors. Other 70 patients were excluded because they were admitted with metastatic disease or they had sarcomas of the trunk or retroperitoneum. In the end, 227 patients met the inclusion criteria of the study.
- Population is under-treated, why did so few patients receive neoadjuvant chemotherapy and radiation therapy?
For the study period (2008-2018), evidence at that time indicated the use of adjuvant radiotherapy for high grade tumors larger than 5 cm or with positive or close borders, while the benefit of chemotherapy was very marginal in disease free survival but not in overall survival in high histologic grade tumors. Thus, only 11.9% of the patients received neoadjuvant radiotherapy, while 37.3% received it as adjuvant treatment. Chemotherapy was administered in only 3.1% of patients in the neoadjuvant setting and in 22.4% as adjuvant treatment. At present, the Multidisciplinary Board od Sarcomas has a greater use of radiotherapy in the neoadjuvant setting; thus, in the Functional Unit, patients with sarcomas of the extremities, especially well-differentiated liposarcomas with borderline resectability and grade 2 and 3 dedifferentiated tumors, are referred to neoadjuvant radiotherapy and subsequent surgical management for limb salvage. In addition, we use the Sarculator nomogram to define the benefit of neoadjuvant chemotherapy in selected cases in a multidisciplinary meeting. This is also described in the article.
- High rate of lymph node involvement, how do you explain this?
Sarcomas have hematogenous dissemination, but there are some histological subtypes, such as rhabdomyosarcoma, clear cell sarcoma, epithelioid sarcoma, synovial sarcoma, and angiosarcoma, that lead to lymph node metastasis. In the Functional Unit, epithelioid sarcomas and angiosarcomas with negative lymph node involvement undergo sentinel lymph node biopsy. This procedure was performed in 8.3% of the patients in the cohort. Lymphadenectomy was performed in 7.9% of the patients who had histologically proven axillary or inguinal metastatic involvement. 11% (n=25) of the patients had synovial sarcomas and 15.4% (n=35) had other histological types, including epithelioid sarcoma, rhabdomyosarcoma, and epithelioid sarcoma. The other explanation is that most of the patients admitted to the INC are patients with a low socioeconomic level and are admitted with very advanced diseases. The discussion session includes an explanation of lymphadenectomy.
- Long term complications are not graded?
It is very challenging due to the difficulty of monitoring these patients. Honestly, we are not very strict in recording the degree of functional limitation of the limb. This is something we are trying to implement.
- In OS analysis paragraph: why does the total % not equal 100%? (Page 7 paragraph 1)
Regarding the OS analysis, at the time of study closure, 37% (n=84) of the patients were alive with no evidence of disease, 26% (n=60) had died from the disease, 18% (n= 40) had died from another cause, 2% (n=5) remained alive with clinical or imaging evidence of the disease, and 38 (17%) did not complete follow-up.
- So there is a discrepancy between text and figure 3: only 33% of patients relapsing, but on the KM curve, it looks like it’s almost 50%?
The text refers to 33% disease-free recurrence and the graph is for overall survival.
- You need tables with number at risk beneath your KM curves
The graphs have been updated
- It is overall hard to understand when the authors are referring to univariate versus multivariate analyses. This should be explained/stated a little better in the results section.
Nowhere in the text we mentioned the univariate analysis,
Round 2
Reviewer 2 Report
Comments and Suggestions for Authors
Some weak points are still present: retrospective study, long time for recruitment, mixed histological types, few patients treated per year.
On the contrary important changes have been made in discussion. The paper is now acceptable to be considered foor pubblication.
Extremely good statistical analysis.